**communications** engineering

# Kinetic inductance current sensor for visible to near-infrared wavelength transition-edge sensor readout
Paul Szypryt [1,2] ✉, Douglas A. Bennett[2], Ian Fogarty Florang [1,2], Joseph W. Fowler [1,2], Andrea Giachero[1,2,3], Ruslan Hummatov [1,2,4], Adriana E. Lita[2], John A. B. Mates[2], Sae Woo Nam[2], Galen C. O'Neil[2], Daniel S. Swetz[2], Joel N. Ullom[1,2], Michael R. Vissers[2], Jordan Wheeler[2] & Jiansong Gao[2,5]

Single-photon detectors based on the superconducting transition-edge sensor are used in a number of visible to near-infrared applications, particularly for photon-number-resolving measurements in quantum information science. To be practical for large-scale spectroscopic imaging or photonic quantum computing applications, the size of visible to near-infrared transition-edge sensor arrays and their associated readouts must be increased from a few pixels to many thousands. In this manuscript, we introduce the kinetic inductance current sensor, a scalable readout technology that exploits the nonlinear kinetic inductance in a superconducting resonator to make sensitive current measurements. Kinetic inductance current sensors can replace superconducting quantum interference devices for many applications because of their ability to measure fast, high slew-rate signals, their compatibility with standard microwave frequency-division multiplexing techniques, and their relatively simple fabrication. Here, we demonstrate the readout of a visible to near-infrared transition-edge sensor using a kinetic inductance current sensor with 3.7 MHz of bandwidth. We measure a readout noise of $1.4\,\mathrm{pA}/\sqrt{\mathrm{Hz}}$, considerably below the detector noise at frequencies of interest, and an energy resolution of $(0.137 \pm 0.001)$ eV at 0.8 eV, comparable to resolutions observed with non-multiplexed superconducting quantum interference device readouts.

The transition-edge sensor (TES)[1,2] has long been used in bolometers[3,4] for sensitive power measurement and calorimeters[5–7] for energy-resolved single-photon detection. At visible to near-infrared (VNIR) wavelengths, the TES calorimeter has a number of advantages over more conventional detectors, such as broad wavelength coverage, intrinsic energy resolution, high quantum efficiency, short (microsecond) photon detection timescales, and negligible dark counts[6,8,9]. These qualities have facilitated the experimental testing of fundamental laws of quantum mechanics, in the setting of loophole-free Bell tests[10–12], and photonic quantum computing[9,13]. They also make the VNIR TES well-suited for exoplanet atmospheric spectroscopy[14,15] and biological imaging[16,17], among other applications. These measurements, however, are limited by the small array sizes currently achieved.

In a TES calorimeter, a superconducting thin film is cooled below its superconducting critical temperature, $T_C$, and then electrically biased into the superconducting-to-normal transition. In this narrow region, the device resistance depends strongly on temperature, causing absorbed photons that

increase the TES temperature to also measurably raise its resistance. Under voltage bias, the TES current response provides a sensitive measurement of an absorbed photon's energy.

Much of the difficulty with TES operation comes in the readout of large arrays. Generally, TESs need to be multiplexed at the cryogenic stage to reduce wiring complexity, thermal loads, and power consumption. In multiplexed readout, signals from many individual TESs are encoded in an orthogonal basis set, measured through a shared cryogenic amplifier chain, and decoded at room temperature. TES currents have historically been read out with superconducting quantum interference devices (SQUIDs)[18] through a variety of multiplexing schemes[19]. The most mature of these is time-division multiplexing (TDM)[20–22], but MHz frequency-division multiplexing (FDM)[23,24] and microwave SQUID multiplexing (μMUX)[25–27] systems have also been implemented. Kilopixel-scale readouts are now typical in long-wavelength bolometers[28–30] and are actively being developed for x-ray calorimeters[31,32]. These readouts have been more challenging to

[1]Department of Physics, University of Colorado Boulder, Boulder, CO, USA. [2]National Institute of Standards and Technology, Boulder, CO, USA. [3]Department of Physics, University of Milano Bicocca, Milan, Italy. [4]Present address: Quantum Design, Inc, San Diego, CA, USA. [5]Present address: AWS Center for Quantum Computing, Pasadena, CA, USA. ✉e-mail: paul.szypryt@nist.gov

realize for the faster VNIR TESs. In TDM, the sampling interval (typically microseconds) is limited by sequential switching of DC-SQUIDs[21,22]. FDM readouts with MHz carrier frequencies are typically limited to 100-kHz-scale signal bandwidths[24]. In µMUX, the flux ramp signal used to linearize the RF-SQUID response limits the sampling rate, and operating a sufficiently fast flux ramp without degrading performance has proved difficult[33,34].

Here, we introduce the kinetic inductance current sensor (KICS), a superconducting resonator-based device that replaces the SQUID in the readout circuit of a TES or other cryogenic detector/device. The KICS, shown schematically in Fig. 1, exploits the nonlinear current dependence of the kinetic inductance in a superconductor. This nonlinearity has previously been explored in other superconducting devices such as the kinetic inductance traveling-wave parametric amplifier (KITWPA)[35–37] and the kinetic inductance parametric up-converter (KPUP)[38,39]. The form of the kinetic inductance nonlinearity can be derived from Ginzburg-Landau theory[40] and expanded[41] as

$$L_{KI}(I) = L_{KI}(0)\left[1 + \frac{I^2}{I_{*,2}^2} + \frac{I^4}{I_{*,4}^4} + \dots\right]. \quad (1)$$

Here, $I$ is the inductor current, and $I_{*,n}$, which is on order the superconducting critical current $I_C$, sets the magnitude of the nonlinearity and depends on the inductor material and cross-sectional area. As the resonant frequency, $f_r$, goes as $1/\sqrt{L(I)\cdot C}$, it can be arbitrarily set by adjusting $I$ up to $I_C$. We define the responsivity as $d|x|/dI$, where $x = \delta f_r/f_r = -\delta L/2L$ is the fractional frequency shift. To leading order, $d|x|/dI \approx I/I_{*,2}^2$. The responsivity, therefore, can be increased by applying a bias current. Readout is done in much the same way as the microwave kinetic inductance detector (MKID)[41–43], another superconducting resonator-based device. Here, a microwave transmission line couples the device to room temperature readout electronics which monitor shifts in the resonant frequency.

Central to the KICS scheme is a superconducting switch, $SW_S$, that is used to trap a persistent current, $I_P$, in the resonator through the requirement of flux quantization. This is analogous to the biasing scheme of

metallic magnetic calorimeters (MMCs)[44] and provides several critical advantages over other microwave domain readouts such as µMUX and KPUPs. First, the persistent current allows the KICS to be self-biased in the high responsivity regime. This prevents pickup in the bias line from entering the KICS circuit and reduces noise when compared to an actively biased device. Additionally, this self-biasing method is dissipationless, reducing power consumption. Finally, superconducting resonator-based detectors and readouts have historically been plagued by fabrication imperfections that cause resonances to deviate from their designed values, creating frequency collisions that reduce device yield[25,45]. The KICS biasing scheme overcomes this issue through continuous frequency tunability[46] and superconducting switches that lock resonances at desired points.

The KICS has further advantages over SQUID readouts. As discussed above, fast detector timescales, such as those observed in VNIR TESs, can make SQUID-based multiplexing difficult. The KICS readout speed, however, is effectively set by the designed resonator bandwidth and can be matched to these fast devices. Furthermore, large and dense arrays are essential for most imaging applications, likely requiring integrated detector and readout fabrication, but SQUIDs have not yet been adapted for this format. For one, fabrication of SQUID readouts can be complex, requiring small and highly uniform junctions and many fabrication steps and materials[25,47]. KICS devices require only a single layer for the critical resonator structures, making high-yield integrated fabrication more feasible. Additionally, KICSs have the potential to be considerably smaller than SQUIDs and thus can be more easily integrated within a TES array. Current is coupled galvanically to a KICS rather than inductively as in a SQUID. As a result, a KICS does not require large coupling coils and associated features that mitigate cross-talk and unwanted backgrounds[48]. SQUID readout cells of >0.4 mm² are typical[9,48,49], whereas the KICS follows many of the design principles of MKIDs for which much more compact designs (0.02 mm²) have already been demonstrated[45].

The KICS can be used to sensitively measure the current from a variety of superconducting devices historically read out with SQUIDs. This includes TES-based detectors operating across the electromagnetic spectrum as well as MMCs. In this manuscript, we demonstrate the KICS concept through the readout of a VNIR TES, which, as noted above, has been especially challenging to multiplex and read out with SQUIDs.

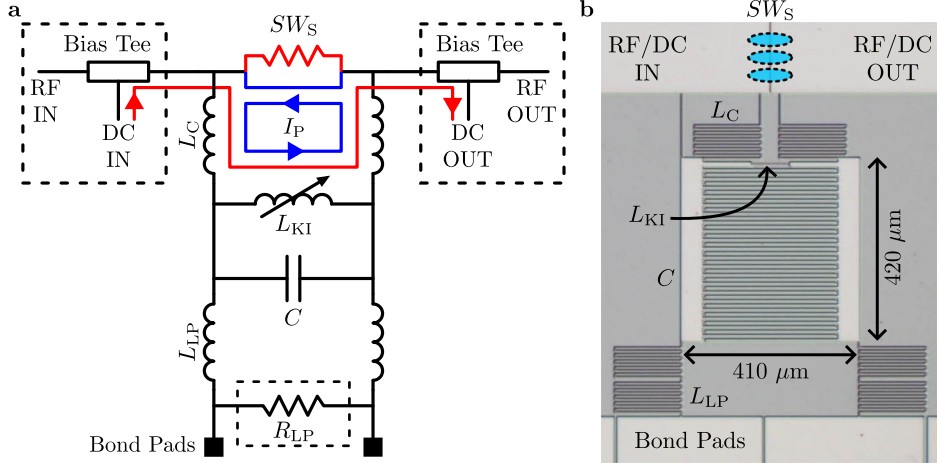

**Fig. 1 | Schematic and micrograph of a NbTiN kinetic inductance current sensor (KICS) implementation.** A 20 nm thick NbTiN layer is used to form an interdigitated capacitor, $C$, and a 0.7 µm wide inductor, $L_{KI}$, that varies nonlinearly with input current. The resonator is inductively coupled to a microstrip transmission line through a set of meandering structures with combined inductance $L_C$. A set of bias tees is used to insert DC current into the nonlinear inductor, and a superconducting switch, $SW_S$, is used to trap this bias current in the device and form a persistent current, $I_P$. In this implementation, $SW_S$ is realized with Al wire bonds closing a gap in the microstrip transmission line, as represented by the blue ovals drawn over the micrograph in (**b**). Near the bottom of the schematic, inductor $L_{LP}$ and resistor $R_{LP}$ form a low-pass filter that separates the microwave circuit of the resonator from that of the lower frequency device (e.g., detector) being read out. Similar to $L_C$, the inductance $L_{LP}$ is formed through a set of inductive meanders, here connecting the resonator to bond pads. The components inside the dashed boxes in (**a**) are not part of the NbTiN KICS chip and are therefore not pictured in (**b**). Instead, the bias tees are assembled onto the device box and $R_{LP}$ is wire bonded into the circuit.

## Results

### Device and setup

The KICS used in this study is adapted from a previous tunable resonator design[46]. The fabrication involves only a single 20 nm thick NbTiN layer with $T_C \approx 14$ K and sheet inductance $L_S \approx 10$ pH/□. This layer was patterned to form the resonator, microstrip transmission line, and coupling and filter inductors. To increase the degree of nonlinearity, the width of the resonator inductor was designed to be small (0.7 μm). In this initial implementation, Al wire bonds ($T_C \approx 1.2$ K) were placed over a gap in the transmission line to form a thermally-activated superconducting switch. A filter resistor, $R_{LP} \approx 60$ mΩ, consisting of a thin layer of Au on Si was wire bonded into the circuit. The dominant inductance in the readout circuit is $L_{LP} \approx 60$ nH, as $L_{KI}$ is on order 1 nH.

The VNIR TES is a 20 μm square trilayer consisting of 2 nm amorphous Si (a-Si), 20 nm W, and 2 nm a-Si, similar to previously reported devices[8,9]. The a-Si is used to stabilize $T_C$[50], here 188 mK. The TES chip was micromachined to match the inner diameter of a zirconia (ZrO$_2$) sleeve, allowing self-alignment of the TES to an optical fiber[51]. As detection efficiency is not the focus of this study, the TES was not embedded in an optical stack previously used to achieve near-unity narrowband efficiency[8]. To voltage-bias the TES, a commercially available 10 mΩ shunt resistor, $R_{sh}$, was connected via wire bonds. The full KICS and TES assembly is shown in Fig. 2.

An adiabatic demagnetization refrigerator (ADR) with base temperature <40 mK was used to cool the device. The cryostat was outfitted with the necessary DC and RF cabling, high-electron mobility transistor (HEMT) amplifiers, and single-mode (SM) fiber to fully interface with the device. Characterization was done using a vector network analyzer (VNA) and microwave homodyne readout, common in MKID measurements[43]. Additional experimental setup details can be found in "Methods".

### Responsivity and noise

To begin characterization of the KICS, the ADR temperature was set to 1.6 K, well below the $T_C$ of the NbTiN resonator but above the $T_C$ of the Al superconducting switch. Here, a DC current, $I_{DC}$, applied at the bias tee is forced to travel around the switch and through the resonator. The VNA was used to sweep the KICS complex transmission as a function of $I_{DC}$ with constant −75 dBm microwave power at the input of the device box.

As shown in Fig. 3a, as $I_{DC}$ was swept upwards, $f_r$ decreases due to increasing $L_{KI}$. For this broad sweep, $f_r$ was calculated at the local minimum of the transmission magnitude. The resonant frequency was measured to be $f_{r,0} = 4.575$ GHz at $I_{DC} = 0$ mA and had a maximal frequency shift of ~700 MHz at $I_C \approx 2.1$ mA. For current values $I_{DC} > I_C$, the nonlinear inductor transitions to the normal state, effectively destroying the resonance. The magnitude of the fractional frequency shift, $|x|$, is plotted against $I_{DC}$ in Fig. 3b. Here, $|x|$ is defined as the shift in frequency from the zero current frequency, scaled by the frequency at the bias point ($f_r = 4.057$ GHz). These data were fit to the $|x|$ form of Eqn. (1) to extract $I_{*2} = (5.11 \pm 0.06)$ mA and $I_{*4} = (2.92 \pm 0.02)$ mA. We note that although higher order terms may be able to reduce some of the systematic offset seen in Fig. 3b, we chose to fit only up to the $I_{*4}$ terms for simpler comparison to other nonlinear kinetic inductance devices reported in the literature.

To prepare the VNIR TES readout, $I_{DC}$ was set to 1.95 mA, where $d|x|/dI = 0.25$ mA$^{-1}$. This bias point was selected to maximize $d|x|/dI$ while still being far enough from $I_C$ to prevent current fluctuations from driving the inductor normal and releasing $I_P$. With the bias set, the ADR temperature was lowered below the $T_C$ of the Al superconducting switch, trapping the bias current in the resonator and establishing $I_P$. At this bias point, $f_r = 4.057$ GHz, a shift of nearly 520 MHz. The zero point of $x$ was moved to this frequency for subsequent measurements, hereafter $x_P$. The resonator bandwidth was measured to be 3.7 MHz ($Q = 1.1 \times 10^3$, largely coupling-limited), sufficient for tracking current changes in the TES with expected microsecond timescales.

After locking in the KICS bias, TES current-voltage (IV) characteristic curves were collected at temperatures between 60 mK and 190 mK, as

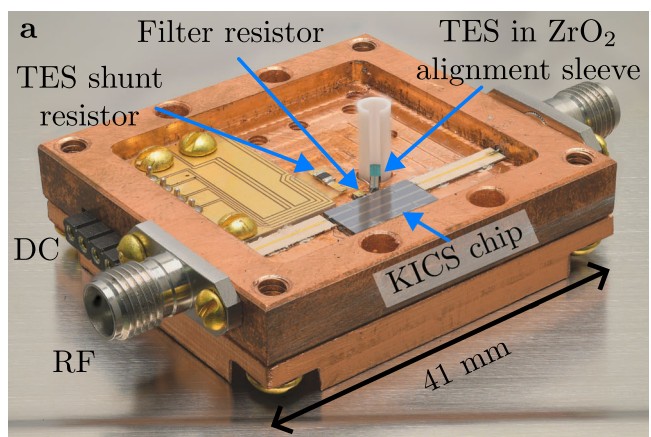

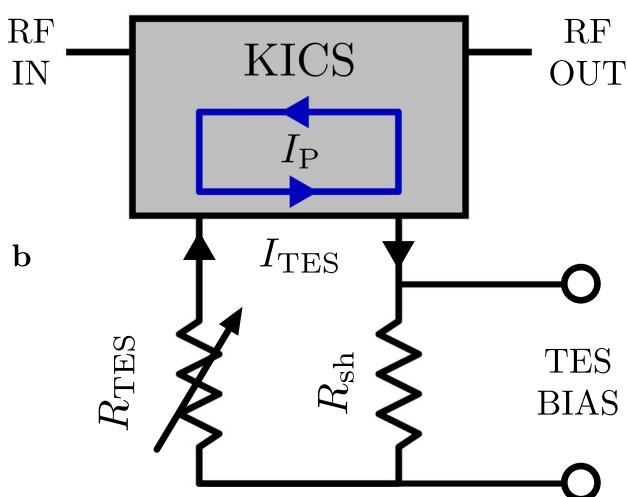

**Fig. 2 | Full kinetic inductance current sensor (KICS) and transition-edge sensor (TES) assembly. a** Shows a photograph of the full device assembly. Here, a visible to near-infrared TES is mounted inside a zirconia (ZrO$_2$) sleeve that is used to align the TES to an optical fiber[51]. Also pictured is the KICS chip and resistors that form the readout circuit. **b** Shows the full device circuit when set up for photon detection. The TES with variable resistance, $R_{TES}$, is biased using a shunt resistor, $R_{sh}$. The TES current, $I_{TES}$, is coupled to a KICS with persistent current $I_P$.

shown in Fig. 4. At each temperature, the TES bias was stepped from high to low, shifting the TES current, $I_{TES}$, and in turn the KICS $f_r$. The measured $d|x|/dI = 0.25$ mA$^{-1}$ was used to convert $x_P$ to $I_{TES}$. An operating temperature of 60 mK was chosen to perform the remainder of TES measurements, although we note that the TES performance (and IV curve) was largely insensitive to operating temperature this far below the TES $T_C$. The IV curve at 60 mK was used to select a TES bias point of 1% $R_N$, where $R_N \approx 10$ Ω is the TES normal state resistance. Generally, a low bias point such as this is preferred to maximize the dynamic range and sensitivity of the TES calorimeter. In practice, a low bias point can cause TES instability (electrothermal runaway) if the electrical timescale, limited by the readout circuit inductance, becomes slow compared to the detector speed[2,7]. In this work, however, the readout circuit inductance was sufficiently small, and instability was not observed even at this low of a TES bias point.

Noise data were collected with a microwave homodyne readout setup sampling at 2.5 MHz. The probe tone was set to the KICS resonance with a power of −70 dBm at the input of the device box, driving the resonator to the onset of bifurcation. Traces in the readout's IQ (in-phase and quadrature) mixer outputs were collected for a total of 30s. Using a frequency sweep of the resonator transmission, these traces were projected onto the frequency and dissipation quadratures (tangent and orthogonal to the

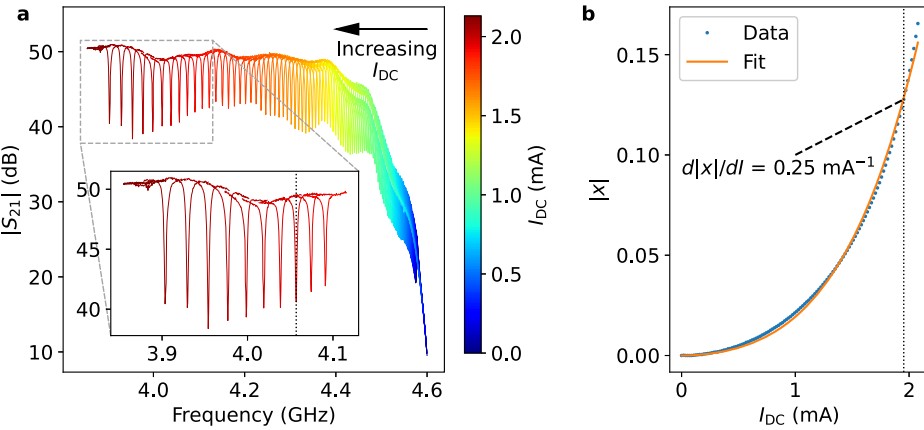

**Fig. 3 | Kinetic inductance current sensor (KICS) responsivity. a** Shows the device transmission magnitude, $|S_{21}|$, as DC current, $I_{DC}$, is swept, with the resonance moving from higher to lower frequencies. The dip in $|S_{21}|$ at high frequencies is due to parasitic capacitance to ground, but this is largely inconsequential for this measurement as increasing $I_{DC}$ to the operating point moves the KICS resonance sufficiently far from the parasitic resonance. **b** Shows the resonator fractional frequency shift magnitude, $|x|$, as a function of $I_{DC}$. The responsivity, $d|x|/dI$, was measured to be 0.25 mA$^{-1}$ at a bias current of 1.95 mA (local slope). This bias point is indicated in both panels with the vertical dashed line.

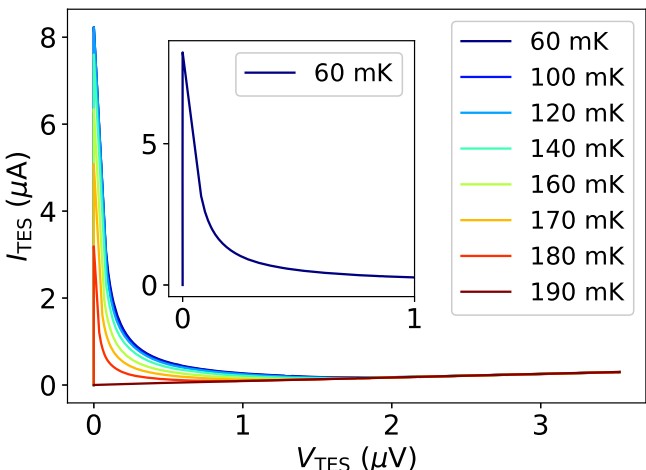

**Fig. 4 | Transition-edge sensor (TES) current-voltage (IV) characteristic curves.** The TES voltage, $V_{TES}$, was swept at temperatures between 60 mK and 190 mK, and the current, $I_{TES}$, was read out through the kinetic inductance current sensor. The inset shows the IV curve at 60 mK, which is the control temperature used for subsequent measurements.

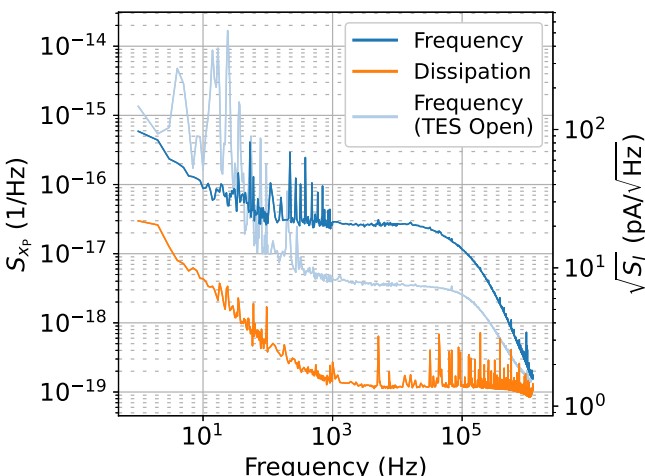

**Fig. 5 | Noise power spectral density (PSD).** The complex transmission data were projected onto the frequency and dissipation quadratures, and Fourier transform techniques were used to generate a noise PSD in each quadrature. The measured responsivity was used to convert the noise PSD in fractional frequency shift units, $S_{x_P}$, to a current sensitivity, $\sqrt{S_I}$. For visual clarity, the frequency resolution was dynamically adjusted from 1 Hz to 1 kHz between the low and high frequency regions. The frequency-quadrature noise was also measured in a separate cooldown with the TES disconnected from the circuit, reflecting the Johnson noise contribution from the filter.

resonance circle in the complex plane) and converted to fractional frequency shift units, $x_P$. Fourier transform techniques were then used to calculate the noise power spectral density (PSD), $S_{x_P}$, up to the 1.25 MHz Nyquist frequency, as shown in Fig. 5. The current sensitivity, $\sqrt{S_I}$, was also calculated using the previously measured value of $d|x|/dI$.

The frequency-quadrature noise is particularly important as this quadrature is most sensitive to the largely dissipationless KICS current response. For the fast TES timescales, only the noise at high frequencies (>kHz) is consequential for detector performance. Here, we measured a mean TES white noise level of 21 pA/$\sqrt{\text{Hz}}$ between 2 kHz and 20 kHz. This includes the contribution from the filter Johnson noise (7.6 pA/$\sqrt{\text{Hz}}$), which was measured in a separate cooldown with the TES disconnected from the circuit. We note that the dataset with the TES disconnected was collecting in an environment with worse low frequency electronics noise, as can been seen in Fig. 5.

The 1/$f$ noise structure observed at low frequencies is expected to largely originate from general electronics noise (quadrature-insensitive) and resonator two-level system (TLS) noise (frequency quadrature)[43]. Above ~1 kHz, the HEMT amplifier provides a noise floor of 1.4 pA/$\sqrt{\text{Hz}}$ (quadrature-insensitive), which is observed in the dissipation quadrature and ultimately limits the readout noise. As can be seen, the readout noise is considerably below that of the TES at frequencies of interest, ensuring readout noise does not affect TES performance.

## Photon characterization

A 1550 nm (0.8 eV) laser diode was used to characterize the VNIR TES photon response. A SM fiber with a 9 µm core diameter (smaller than TES lateral dimensions) was used to efficiently couple light to the TES. The laser diode was pulsed with a 30 ns width (short compared to TES timescales) and 80 Hz repetition rate to generate a total of 10,000 low-photon-number pulses. This value was chosen to ensure sufficient counts for generating a peak-resolved energy spectrum (below).

Analogous to the noise data processing, optical pulse events were read out with the homodyne readout sampling at 2.5 MHz, and the I and Q traces were converted to and analyzed in the frequency quadrature. Pulse records processed in this manner are shown in Fig. 6a, with the pulse height characteristic of the absorbed energy. Here, the traces are clearly separated into discrete groups representing varying numbers of 1550 nm photons, demonstrating the photon-number-resolving capability of the TES paired with the KICS. An average 1/e fall time of 4.7 µs was measured for the single-photon pulses.

We used the Microcalorimeter Analysis Software Suite (MASS)[52] to further process the pulse records[53]. Here, the traces were optimally filtered[54]

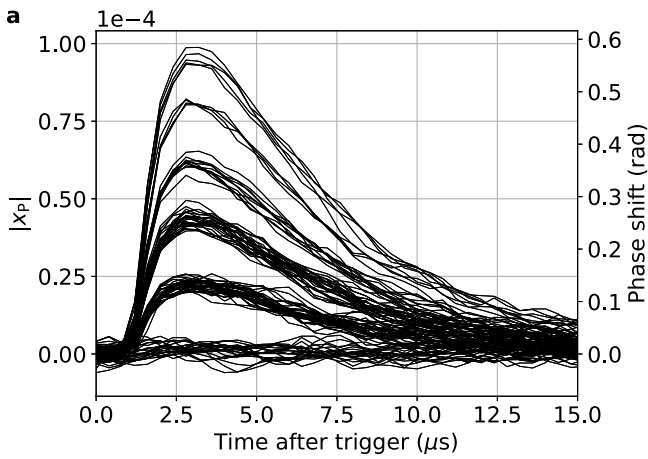

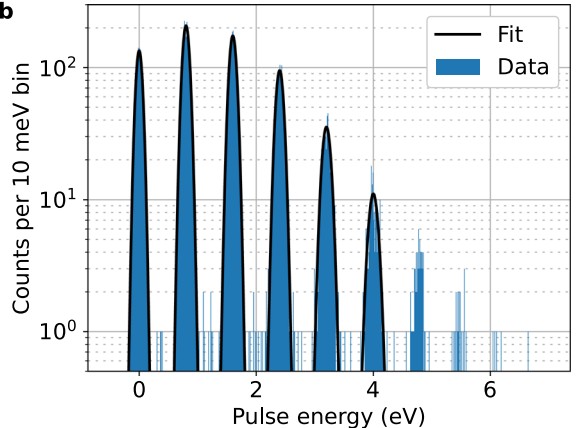

**Fig. 6 | Response to 1550 nm photons. a** Shows the fractional frequency shift response, $|x_P|$, to the first 100 1550 nm photon pulses, with clear separation by photon number. A total of 10,000 traces were filtered and energy calibrated to produce the spectrum in (**b**). The peaks in the spectrum were fit to Gaussian distributions with the extracted width representing the energy resolution, $\Delta E$, at that peak energy (see Table 1).

**Table 1 | Energy resolution and resolving power as function of absorbed photon energy**

| *E* (eV) | $\Delta E$ (eV) | *R* |
|---|---|---|
| 0.0 | 0.127 ± 0.002 | N/A |
| 0.8 | 0.137 ± 0.001 | 5.84 ± 0.05 |
| 1.6 | 0.142 ± 0.001 | 11.3 ± 0.1 |
| 2.4 | 0.148 ± 0.003 | 16.2 ± 0.3 |
| 3.2 | 0.168 ± 0.007 | 19.0 ± 0.8 |
| 4.0 | 0.183 ± 0.024 | 21.9 ± 2.9 |

large $d|x|/dI = 0.25 \, \text{mA}^{-1}$ was chosen to enable high sensitivity to small TES current changes. We also demonstrated KICS self-biasing through a persistent current, useful for reducing pickup from the bias line and readout noise.

With the KICS self-biased, we characterized the VNIR TES readout through IV curve and noise measurements. We observed the readout noise at frequencies of interest ($1.4 \, \text{pA}/\sqrt{\text{Hz}}$) to be HEMT amplifier limited and considerably below the TES (and filter) noise ($21 \, \text{pA}/\sqrt{\text{Hz}}$). We then characterized the TES response to 1550 nm photon pulses and measured a single-photon energy resolution of ($0.137 \pm 0.001$) eV, a value already comparable with DC-SQUID readout. We note that the filter was not well-optimized and added a small but non-negligible amount of noise, but this could be improved with larger $R_{LP}$. Additionally, the readout noise floor could potentially be improved by using KITWPAs instead of HEMT amplifiers. While this is an active area of research, KITWPAs can theoretically achieve quantum-limited noise performance, whereas their HEMT amplifier counterparts typically achieve noise levels of at least 10 times above the standard quantum limit[35,37].

Next steps include demonstrating multiplexable readout through a small VNIR TES array read out with multiple KICS devices coupled to a common microwave transmission line. Due to the resonator-based nature of the KICS, we expect this to closely resemble the microwave readouts of MKID arrays[58] and do not anticipate major obstacles. The unique application of persistent currents, however, will need to be further studied to determine the impact on crosstalk between readout channels. Following the readout demonstration, we plan to develop integrated KICS and TES arrays with a targeted 100-μm-scale pixel pitch. Although the prototype KICS device contained relatively large features for ease of fabrication, 100-μm-scale devices could likely be achieved by following many of the same design principles as MKIDs[45]. Integrated fabrication will only add a small number of layers to the current TES fabrication process, mainly for the KICS resonators, superconducting switches, and TES shunt and filter resistors. Integrated fabrication will likely be necessary for realizing the dense, kilopixel-scale arrays needed for many astronomy applications, such as exoplanet direct imaging and atmosphere spectroscopy.

As arrays get larger, it would also be useful to individually bias KICS devices in order to tune resonance locations, preventing frequency collisions that reduce yield. In this direction, preliminary work has begun on superconducting bilayer switches fabricated with a thickness ratio and $T_C$ gradient, giving each device a unique temperature at which an optimal $I_P$ can be trapped. Other methods are likely possible though may require increased wiring complexity.

When looking toward large arrays, it is useful to compare the potential multiplexing capability and effective readout channel bandwidth of KICS devices to their SQUID counterparts. Assuming a typical 4 GHz to 8 GHz microwave readout band[27,45], signal bandwidth of half the resonator linewidth, and 8 linewidth spacing to minimize crosstalk, the KICS readout has an effective bandwidth of 250 MHz on a single readout line. SQUID-based μMUX readouts have similar effective bandwidths, although here the bandwidth is further reduced by a factor of two from typical SQUID modulation, resulting in an effective bandwidth of 125 MHz. TDM and FDM readouts have historically been limited to effective bandwidths of

to maximize the signal-to-noise ratio of the pulse energy estimation. The average pulse trace was combined with photonless noise data to generate the optimal (Wiener) filter. Filtered values representative of the pulse energy were then computed by taking the dot product of each pulse trace with the filter.

The filtered values were energy calibrated using a linear interpolation function generated with the known energies of the first six peaks in the filtered value histogram (zero to five 1550 nm photons). The energy-calibrated spectrum, shown in Fig. 6b, is consistent with a Poisson distribution with $\lambda = 1.7$. A Gaussian fit was used to extract a FWHM energy resolution $\Delta E = (0.137 \pm 0.001)$ eV at the 0.8 eV single-photon peak (resolving power $R = E/\Delta E = 5.84 \pm 0.05$), comparable to resolutions of similar detectors read out with non-multiplexed DC-SQUIDs[9]. A comparison of $\Delta E$ up to the 5-photon peak is shown in Table 1. As can be seen, the TES is fairly linear ($\Delta E$ is roughly flat) across this energy range, but compression begins to degrade $\Delta E$ at the higher photon-number peaks.

## Discussion

Here, we introduced the principles of the KICS and demonstrated its viability for VNIR TES readout. The kinetic inductance nonlinearity of the KICS largely behaves according to Eqn. (1), and the current sets the responsivity, $d|x|/dI$, and with it the dynamic range of the readout. This is analogous to the inductive coupling (input mutual inductance) in SQUID readouts[55–57], although we note with the KICS this can be arbitrarily tuned during cryogenic operation rather than be fixed at fabrication. In this work, a

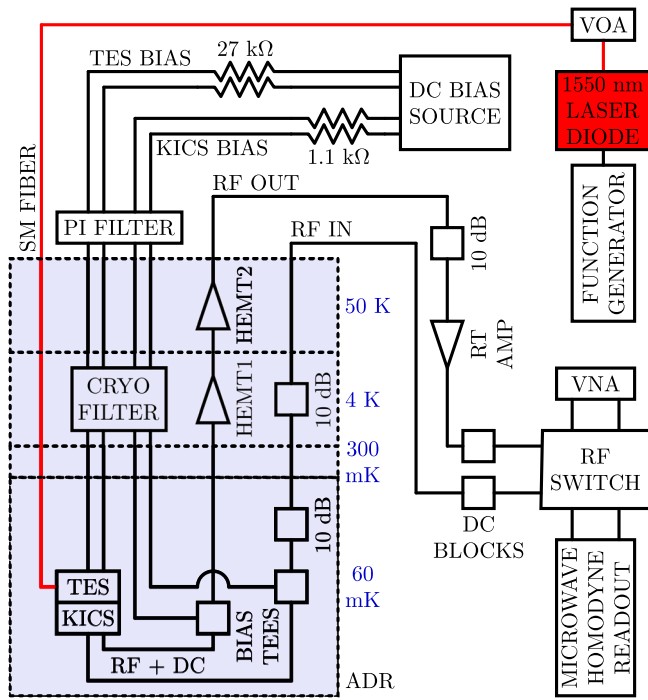

**Fig. 7 | Experimental setup schematic.** The setup is separated into optical, DC, and RF circuits. The optical circuit is used to source single photons to the TES detector through a single-mode (SM) fiber, and photon pulses are shaped using a function generator and variable optical attenuator (VOA). The DC circuit is used to bias the transition-edge sensor (TES) and kinetic inductance current sensor (KICS). Characterization measurements are done with the RF circuit through a vector network analyzer (VNA) and microwave homodyne readout. The RF signals are amplified using a set of cryogenic high-electron mobility transistor (HEMT) amplifiers and a room temperature amplifier (RT amp). The shaded region represents the interior of the adiabatic demagnetization refrigerator (ADR), separated by temperature stage.

10 MHz or less[21–24]. For an example TES detector with 250 kHz signal bandwidth, this results in multiplexing factors of 1000 for KICS readout, 500 for μMUX, and 40 or less for TDM and FDM.

Our work points to another advantage of KICS readout: compatibility with very low circuit inductances. Finite inductance in a TES bias circuit limits TES speed due to the onset of electrothermal instability when the electrical and thermal time constants converge. Here, the circuit inductance was dominated by our conservative choice of $L_{LP} \approx 60$ nH, but the limit on inductance is set by $L_{KI}$ which was ~1 nH. In contrast, the use of flux coupling in SQUID circuits necessitates higher inductances in the TES circuit to achieve comparable levels of current noise. Low inductance KICS circuits are particularly well matched to the readout of TESs with sub-microsecond recovery times for continuous-variable photonic quantum computing.

In this work, we have demonstrated for the first time a viable alternative readout technology for VNIR TES devices, with a clear path to multiplexed array readout. The KICS could also be extended to other wavelengths and detector technologies. In particular, TES-based x-ray/gamma-ray calorimeters tend to operate at slower timescales than the VNIR TES devices described here. Although SQUIDs provide sufficient readout bandwidth here, the KICS represents a simpler and more scalable alternative that could make large array readout more feasible. Furthermore, the KICS could be used for the readout of MMCs[44], which also use a self-biasing scheme and have historically been limited by the bandwidth of multiplexed SQUID readouts and the noise floor of HEMT amplifiers. Here, the KICS could be especially impactful, particularly if integrated with a quantum-noise-limited KITWPA that shares the same NbTiN material system. Finally, the KICS is not necessarily limited to superconducting detector readout, and any other cryogenic devices

requiring sensitive current readout, currently done using a SQUID, could likely be adapted to the KICS.

## Methods
### Experimental setup overview
A schematic of the overall experimental setup is shown in Fig. 7. An ADR backed by a He-3 sorption cooler was used to perform the measurements, with the KICS and VNIR TES typically controlled at 60 mK. The cryostat contains 4 main temperature stages, with temperatures of 60 mK, 300 mK, 4 K, and 50 K. The experimental setup can roughly be divided into three categories/circuits: optical, DC, and RF. These are described in more detail below.

### Optical circuit
Optical pulses are created by a 1550 nm laser diode actuated with a pulse mode function generator. At each repetition, a single, short-duration voltage pulse is generated, enabling nanosecond-scale photon pulses. The optical output of the laser diode is connected to a variable optical attenuator (VOA), allowing control of the mean photon number in optical pulses. Single-mode (SM) fiber with a core diameter of 9 μm is used to couple the output of the VOA to the VNIR TES inside the cryostat. At each of the temperature stages, the optical fiber is heat sunk to the stage and coiled to filter black-body radiation that couples in from the room temperature side of the fiber. The fiber is terminated with a ferrule and connected to a $ZrO_2$ mating split sleeve. The TES chip is micromachined into a circular shape with diameter matching the $ZrO_2$ sleeve inner diameter and mounted on a sapphire rod of the same diameter within the device box. The $ZrO_2$ sleeve containing the fiber ferrule is inserted over the TES and sapphire rod, self-aligning the TES to the fiber core. The TES uses Nb leads that extend out through the $ZrO_2$ split sleeve gap, enabling wire bonding to the device.

### DC circuit
A multi-channel digital-to-analog converter (DAC) was used as the DC bias source for the KICS (generating persistent current) and TES. Bias resistors of 1.1 kΩ and 27 kΩ were placed at the DAC outputs to be able to source appropriate current levels for the KICS and TES, respectively. The DC bias was applied differentially and symmetrically, and twisted pair wiring was used to mitigate electronics noise and pickup. To further reduce noise, the DC lines were low-pass filtered at three locations. This includes a 10 Hz filter internal to the DAC, a 4 nF pi filter (800 kHz cut-off) at the cryostat feedthrough, and cryogenic filter banks (65 kHz cut-off for lowest frequency bank) at the 4 K stage. To couple DC current through the RF microstrip line and into the KICS resonator, a set of bias tees on the 60 mK stage was used.

### RF circuit
The RF circuit contains the hardware needed to drive and read out the KICS resonator, connected via coaxial cabling. The RF input to the cryostat consists of 10 dB attenuators on the 4 K and 60 mK stages, used to reduce the noise of the room temperature probe tone before reaching the 60 mK device input. The RF output from the device is first amplified at the 4 K stage by a commercial HEMT amplifier (HEMT1) with a specified noise temperature $T_N = 3$ K and gain of 25 dB at 4 GHz. The output is further amplified at the 50 K stage by a second HEMT amplifier (HEMT2). Outside the cryostat, a 10 dB attenuator and a room temperature amplifier are used to adapt to power levels that are suitable for the room temperature readout electronics. DC inner/outer blocks are used to isolate the cryostat and electronics low frequency signals and grounds. Device characterization measurements are performed using a commercial VNA and a custom microwave homodyne readout system, with a RF switch used to choose between the two.

### Microwave homodyne readout
A schematic of the microwave homodyne readout is shown in Fig. 8. A RF synthesizer is used to generate a probe tone at the KICS resonant frequency. A directional coupler is used to split this waveform, with one portion sent to the local oscillator (LO) port of an IQ (in-phase and quadrature) mixer. The

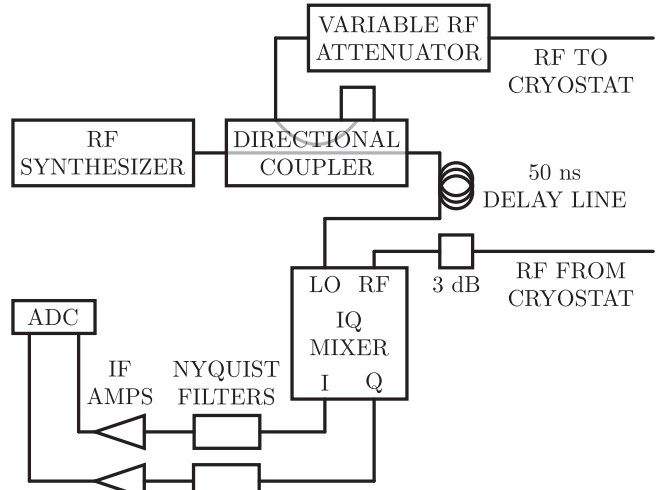

**Fig. 8 | Microwave homodyne readout schematic.** Central to the readout is an IQ (in-phase and quadrature) mixer, which transforms the microwave frequency response of the kinetic inductance current sensor (KICS) into baseband I and Q signals through mixing with a reference signal at the local oscillator (LO) port. These I and Q signals are Nyquist filtered and amplified with intermediate frequency amplifiers (IF amps) before being digitized by an analog-to-digital converter (ADC). The digitized I and Q outputs are used to characterize the device noise and photon response.

other portion is sent through a variable RF attenuator and the KICS device in the cryostat before arriving at the RF port of the IQ mixer. A 50 ns delay line is added in front of the LO port in order to mitigate effects of the cable delay in the I and Q outputs. The I and Q outputs are filtered with 1 MHz cut-off Nyquist filters. Intermediate frequency (IF) amplifiers are used to amplify the I and Q signals to utilize the full range of an analog-to-digital converter (ADC). This 14-bit ADC is used to digitize the I and Q signals at a 2.5 MHz sampling rate.

## Data availability
The data used to generate Figs. 3, 4, 5, and 6 are available via Figshare[59]. All other data used in this study are available from the corresponding author upon reasonable request.

## Code availability
The Microcalorimeter Analysis Software System (MASS), Version 0.8.0, code used to analyze the data in this work is available via Zenodo[52]. All other code is available from the corresponding author upon reasonable request.

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

## Acknowledgements

This work was supported by the NASA APRA program under Grant No. NNH23OB118A. AG is supported by the European Union's H2020-MSCA under Grant No. 101027746. Certain commercial equipment, instruments, or materials are identified in this paper in order to specify the experimental procedure adequately. Such identification is not intended to imply recommendation or endorsement by the National Institute of Standards and Technology, nor is it intended to imply that the materials or equipment identified are necessarily the best available for the purpose.

## Author contributions

P.S., D.A.B, S.W.N., G.C.O., D.S.S., J.N.U., M.R.V., and J.G. conceptualized and developed the work. A.G., A.E.L., and J.G. simulated and designed the KICS and TES devices. M.R.V and A.E.L fabricated the KICS and TES devices. P.S., J.W.F., R.H., G.C.O., J.W., and J.G. developed the experimental hardware and analysis tools. P.S. and I.F.F. conducted the experiments. P.S. and J.A.B.M. analyzed the data. P.S. wrote the manuscript. All authors reviewed the manuscript. One of the contributing authors, S.W.N. has passed away during the preparation of the manuscript. The author's family approved inclusion of their name in the authors list.

## Competing interests

The authors declare no competing interests.
