## [Transparent Peer Review file · Communications Engineering]

Kinetic inductance current sensor for visible to near-infrared wavelength transition-edge sensor readout

Corresponding Author: Dr Paul Szypryt

Version 0:

Reviewer comments:

Reviewer #1

(Remarks to the Author)

The paper presents a kinetic inductance current sensor (KICS) device for reading out the TES with information about design, fabrication, and characterization of the system, including Noise power spectral density (PSD). The nonlinear kinetic inductance is used for tuning the resonator frequency by a persistence current that trapped in a superconducting loop, which is a clever idea. The paper is very interesting, it presents a new and important results for the development of readout methods for TES arrays. I suggest it to be published. Some minor comments are listed below.

1. On page 245, the authors mention that the TES was biased at very low resistance but better energy resolution was obtained. What is the reason? If it is caused by the readout circuit, could it be resolved so that the TES can be biased higher and the overall energy resolution can be improved? A circuit simulation may help.
2. On page 285, is the laser operated in a burst mode for generating 10,000 pulses? Is this the maximum duration that the readout circuit can process?
3. What is the detection efficiency of the TES?

Reviewer #2

(Remarks to the Author)

This paper proposes a novel method to read out Transition Edge Sensors (TES) by making use of a superconducting resonator as a highly sensitive current sensor through its non-linear kinetic inductance. The authors describe the principle of operation and demonstrate a single TES pixel with readout resonator. The presented measurements show TES limited operation (noise, energy resolution). This novel readout method is particularly important for TES in visible & near-infrared wavelengths, but could be a major step forward for the simplification of TES and magnetic microcalorimeters readout and multiplexing in general. The reviewer would also like to commend the authors on a very well written manuscript. Hence it is easy for me to recommend this manuscript for publication.

I do have some questions and minor suggestions to the author to improve the manuscript.

- 1) Figure 1 – I found this figure particularly hard to parse. It required a similar figure from reference [45] and multiple reads of the figure caption and main text to identify each component in the readout resonator in (b) and its matching lumped-element component (a). Using the matching figure of [45] as a guide, I would recommend using coloration and additional annotation (incl. e.g. hand drawing in the wire bonds) for (b) and a clearer notation / box in (a) indicating which elements are / are not shown in (b). I would also note that as currently presented, the “Measurement Circuit Bond Pads” disappears into the bold part of the caption. (A little more white space might be sufficient to mitigate this.)
- 2) Page 4 (Responsivity & Noise) – At which microwave readout power was the VNA measurement taken in relation the resonator’s bifurcation power? Was the absolute power kept constant throughout the measurement or was the power w.r.t. bifurcation kept constant.
- 3) Page 4/5 (Responsivity & Noise) & Fig. 2 – “sudden decline of the resonator Q” – what is the physical origin of this sudden onset of loss?
- 4) Figure 2 (right panel) – a systematic offset is observed between the measurement and fit (fit is below the data at $I_{dc} < 1.5$ (mA) & $I_{dc} > 2.0$ (mA) and above the curve between these values.) Is this systematic offset the result of neglected higher order I/I^* terms or a more complex behavior of the resonator that is not captured by the physical model used?
- 5) Page 6 (first paragraph) – “Although this bias point is ... without cause TES instability” – one or two references would be appreciated to support this or at least give a more diverse audience the chance to read up on TES bias selection and the

TES instability. One or two sentence on potential drawbacks of the low bias point and/or why a higher bias point is usually selected could be optionally included.

6) Page 7 (paragraph on energy resolution; last paragraph before Discussion) – it would be useful to very explicitly define the nomenclature / symbolism for energy resolution here. The paper uses energy resolution for dE, while in astrophysics resolution is used for dE/E. In Fig 6c table dE/E is clearly not flat, while dE is. Slightly more purposeful wording should mitigate this confusion.

7) Page 7 (Discussion) – “inductive coupling in SQUID readouts [...] fixed at fabrication”. A reference would be appreciated to support a more diverse audience.

8) Introduction/Discussion – Currently, the potential for reducing the number of wires to cryogenic temperatures of the KICS is only presented in a qualitative way. Given the breath of specialisms within the authors, they would be in a unique position to quantify and compare the actual wiring needed for a large-array (e.g. 1k) readout using the various TES readouts (TDM, FDM, uMUX, KICS), giving a more balanced judgement on the improvement given the various RF & DC lines these various readouts need for operation.

Reviewer #3

(Remarks to the Author)

The paper describes kinetic inductance current sensor (KICS) for VNIR TES readout. The technology developed by the authors is one promising solution for further large-scale imaging application. The paper describes the design and characterization of the device. The results described in paper will be useful to sensitive current readout.

Comments 1: For large-scale imaging, TES arrays contain many thousands pixels, and the signals need to be read out with many current sensors. In Fig. 1. the size of the KICS including LC, LKI and LLP is about 0.6 mm * 0.6 mm, it could be reduced by optimizing design and fabrication process?

Comments 2: In Fig.5, the noise level is 1.4 pA/Hz^{1/2}, and 1/f corner frequency is about 1 kHz. Comparing with SQUIDs, the noise performance of KICS could be improved by optimization of design and fabrication? The frequency spectrum spans the frequency range of about 1 Hz to 1 MHz in Fig.5, however, in the abstract the bandwidth of KICS is 3.7 MHz, please comment it.

Version 1:

Reviewer comments:

Reviewer #1

(Remarks to the Author)

The authors have responded all my concerns. I would suggest it to be accepted.

Reviewer #2

(Remarks to the Author)

This paper proposes a novel method to read out Transition Edge Sensors (TES) by making use of a superconducting resonator as a highly sensitive current sensor through its non-linear kinetic inductance. The authors describe the principle of operation and demonstrate a single TES pixel with readout resonator. The presented measurements show TES limited operation (noise, energy resolution). This novel readout method is particularly important for TES in visible & near-infrared wavelengths, but could be a major step forward for the simplification of TES and magnetic microcalorimeters readout and multiplexing.

The reviewer would also like to commend the authors on a very well written manuscript and the way the reviewer comments have been incorporated in the new manuscript. In particular Fig. 1 has significantly improved and the formulation around the TES bias point has been made more accessible for a wider audience.

No further questions/comments that required attention are found in this revision.
Hence it is easy for me to recommend this manuscript for publication at this stage.

Reviewer #3

(Remarks to the Author)

In this paper, the authors investigated the kinetic inductance current sensor to make sensitive current measurements and demonstrated its functionality with low noise level, it may be useful for large-scale spectroscopic imaging or photonic quantum computing applications. The submission has been improved and is worthy of publication.

We thank the reviewers for their thoughtful comments and appreciate the time and effort spent in reviewing our manuscript. We believe these comments were instrumental to improving the manuscript, and as such we have followed all reviewer recommendations and revised the manuscript accordingly. Below, we copy the reviewer comments in blue and provide our point-by-point responses in black.

Reviewers' comments:

Reviewer #1 (Remarks to the Author):

The paper presents a kinetic inductance current sensor (KICS) device for reading out the TES with information about design, fabrication, and characterization of the system, including Noise power spectral density (PSD). The nonlinear kinetic inductance is used for tuning the resonator frequency by a persistence current that trapped in a superconducting loop, which is a clever idea. The paper is very interesting, it presents a new and important results for the development of readout methods for TES arrays. I suggest it to be published. Some minor comments are listed below.

1. On page 245, the authors mention that the TES was biased at very low resistance but better energy resolution was obtained. What is the reason? If it is caused by the readout circuit, could it be resolved so that the TES can be biased higher and the overall energy resolution can be improved? A circuit simulation may help.

Typically, we expect to see improved dynamic range and sensitivity (resolution) as the bias point is decreased. At the same time, care must be taken to ensure that as the bias point is lowered, the readout timescale does not become slow compared to the detector speed, which could cause the TES to become unstable through runaway electrothermal oscillations. We have expanded this portion of the text to better explain these tradeoffs. Additionally, we have added the following references, which go over the math behind TES biasing and stability criteria.

https://doi.org/10.1007/10933596_3

<https://doi.org/10.1088/0953-2048/28/8/084003>

As for a circuit simulation, this is an active area of research in the TES community, and unfortunately there are currently no models that fully/accurately capture the complex physics of the TES superconducting-to-normal transition.

2. On page 285, is the laser operated in a burst mode for generating 10,000 pulses? Is this the maximum duration that the readout circuit can process?

The laser diode was driven with a function generator operating in pulse mode at a constant 80 Hz repetition rate. Here, by pulse we mean a single sharp voltage rise and fall rather than a burst of changing voltages. We added some clarification of this to the second paragraph of the Methods section.

The number of pulses was chosen largely arbitrarily but ensured we had sufficient statistics in the linear range of the detector. It is not a limitation of the readout circuit (although unoptimized data acquisition software may have set some nonfundamental limitations). We added a quick statement about this in the Photon Characterization subsection.

3. What is the detection efficiency of the TES?

In this work we were largely interested in characterizing the new KICS readout devices and less interested in detection efficiency. As such, the W TES was not embedded in an optical stack and the detection efficiency is expected to be quite low (similar to 1550 nm absorption of bulk W). As absolute efficiency measurements of cryogenic detectors can be very time-consuming and challenging to implement (requiring additional, well-calibrated hardware), and as the efficiency was not a focus of this work, these measurements were not done. In any case, we modified the text in the Device and Setup subsection to more explicitly state that an optical stack was not used in this study and the detection efficiency is expected to be low.

Reviewer #2 (Remarks to the Author):

This paper proposes a novel method to read out Transition Edge Sensors (TES) by making use of a superconducting resonator as a highly sensitive current sensor through its non-linear kinetic inductance. The authors describe the principle of operation and demonstrate a single TES pixel with readout resonator. The presented measurements show TES limited operation (noise, energy resolution). This novel readout method is particularly important for TES in visible & near-infrared wavelengths, but could be a major step forward for the simplification of TES and magnetic microcalorimeters readout and multiplexing in general. The reviewer would also like to commend the authors on a very well written manuscript.

Hence it is easy for me to recommend this manuscript for publication.

I do have some questions and minor suggestions to the author to improve the manuscript.

1) Figure 1 – I found this figure particularly hard to parse. It required a similar figure from reference [45] and multiple reads of the figure caption and main text to identify each component in the readout resonator in (b) and its matching lumped-element component (a). Using the matching figure of [45] as a guide, I would recommend using coloration and additional annotation (incl. e.g. hand drawing in the wire bonds) for (b) and a clearer notation / box in (a) indicating which elements are / are not shown in (b). I would also note that as currently presented, the “Measurement Circuit Bond Pads” disappears into the bold part of the caption. (A little more white space might be sufficient to mitigate this.)

We have reworked Fig 1 with the goal of making it easier to parse and convey the KICS concept to the reader. In particular, we have made the circuit component labels more consistent between two figure panels. Additionally, we have drawn dashed boxes around the components in panel **a** that do not appear in panel **b**. We have also added ovals to panel **b** to indicate the location of the wire bonds that are added to the circuit to realize the superconducting switch. Finally, we have updated the caption to account for the changes made to the figure. In the caption we now also describe how inductances L_C and L_{LP} each are composed of two lumped components mirrored across the device.

2) Page 4 (Responsivity & Noise) – At which microwave readout power was the VNA measurement taken in relation the resonator’s bifurcation power? Was the absolute power kept constant throughout the measurement or was the power w.r.t. bifurcation kept constant.

The VNA was largely just used to track changes in resonance as a function of some relatively slowly varying sweep parameter (e.g. KICS DC current in Fig 3 or TES bias voltage in Fig 4). Here, the absolute power was kept constant at -75 dBm at the input of the device box. We note that the exact microwave readout power was less important here. We simply wanted to use a value that was high enough so that the resonant frequency estimate is not readout noise limited and low enough so that the microwave power itself does not shift the resonance. In any case, we noted this microwave power at the start of the Responsivity and Noise subsection.

3) Page 4/5 (Responsivity & Noise) & Fig. 2 – “sudden decline of the resonator Q” – what is the physical origin of this sudden onset of loss?

The idea here is that as the DC current in the inductor surpasses the critical current, it becomes normal (lossy). We agree that our explanation of this was somewhat confusing in the original manuscript and have therefore reworded it in the new version.

4) Figure 2 (right panel) – a systematic offset is observed between the measurement and fit (fit is below the data at $I_{dc} < 1.5$ (mA) & $I_{dc} > 2.0$ (mA) and above the curve between these values.) Is this systematic offset the result of neglected higher order I/I^ terms or a more complex behavior of the resonator that is not captured by the physical model used?*

Yes, we found we could get a marginally better fit when including higher order terms, but for simplicity's sake decided to report only up to the I^4 term (which already fits the data reasonably well for the range of currents tested). We felt that this would also make it easier to compare to published results of other nonlinear kinetic inductance devices, which typically only fit the nonlinearity up to the I^2 or I^4 term. We added this clarification to the Fig 3 caption.

5) Page 6 (first paragraph) – “Although this bias point is ... without cause TES instability” – one or two references would be appreciated to support this or at least give a more diverse audience the chance to read up on TES bias selection and the TES instability. One or two sentence on potential drawbacks of the low bias point and/or why a higher bias point is usually selected could be optionally included.

We have expanded the description of this in the text (please see response to Comment 1 of Reviewer 1). Additionally, we have included the following references as part of this discussion:

https://doi.org/10.1007/10933596_3

<https://doi.org/10.1088/0953-2048/28/8/084003>

6) Page 7 (paragraph on energy resolution; last paragraph before Discussion) – it would be useful to very explicitly define the nomenclature / symbolism for energy resolution here. The paper uses energy resolution for dE , while in astrophysics resolution is used for dE/E . In Fig 6c table dE/E is clearly not flat, while dE is. Slightly more purposeful wording should mitigate this confusion.

Here, we had meant for dE to be defined as the energy resolution and $R=dE/E$ to be defined as the energy resolving power. We have changed to text in this paragraph and in the Fig 6 caption to make these definitions more explicitly.

7) Page 7 (Discussion) – “inductive coupling in SQUID readouts [...] fixed at fabrication”. A reference would be appreciated to support a more diverse audience.

We have added the following references discussing configuring the inductive coupling (input mutual inductance) in SQUID readouts of TES devices:

<https://doi.org/10.1007/s10909-012-0509-7>

<https://doi.org/10.1063/1.4754630>

<https://doi.org/10.1117/1.JATIS.5.2.021007>

8) Introduction/Discussion – Currently, the potential for reducing the number of wires to cryogenic temperatures of the KICS is only presented in a qualitative way. Given the breadth of specialisms within the authors, they would be in a unique position to quantify and compare the actual wiring needed for a large-array (e.g. 1k) readout using the various TES readouts (TDM, FDM, uMUX, KICS), giving a more balanced judgement on the improvement given the various RF & DC lines these various readouts need for operation.

We have added a paragraph to the Discussion section to address this comment. Here, we have limited our description to predicted multiplexing factors and effective readout line bandwidths, which we believe is particularly useful for comparing the various TES readout technologies. Although a more extensive cryogenic wiring description than the one we have added may be instructive, we feel that it would unbalance the Discussion section and broader manuscript.

Reviewer #3 (Remarks to the Author):

The paper describes kinetic inductance current sensor (KICS) for VNIR TES readout. The technology developed by the authors is one promising solution for further large-scale imaging application. The paper describes the design and characterization of the device. The results described in paper will be useful to sensitive current readout.

*Comments 1: For large-scale imaging, TES arrays contain many thousands pixels, and the signals need to be read out with many current sensors. In Fig. 1. the size of the KICS including LC, LKI and LLP is about 0.6 mm * 0.6 mm, it could be reduced by optimizing design and fabrication process?*

Yes! Many of the features in the prototype device were larger than necessary for ease of fabrication and quick device testing turnaround. We believe that ultimately, the size of KICS devices could approach sizes currently achieved by VNIR MKIDs, with pixel pitch between 100 um to 200 um. We have added a statement to the Discussion section regarding device scaling.

Comments 2: In Fig.5, the noise level is 1.4 pA/Hz^{1/2}, and 1/f corner frequency is about 1 kHz. Comparing with SQUIDs, the noise performance of KICS could be improved by optimization of design and fabrication? The frequency spectrum spans the frequency range of about 1 Hz to 1 MHz in Fig.5, however, in the abstract the bandwidth of KICS is 3.7 MHz, please comment it.

The low frequency 1/f noise largely consists of room temperature electronics noise and resonator two level system (TLS) noise. TLS noise in resonators is an active area of research, but this could potentially be improved through a better understanding of TLS physics to guide resonator design, more thorough surface/interface cleaning, etc.

In any case, for the application of single photon detection where detector timescales are expected to be fast, improvements to the high frequency end of the spectrum would be most important. This flat portion of the noise PSD is currently limited by HEMT amplifier noise. The use of quantum-limited parametric amplifiers, which can have a noise floor more than 10 times lower than HEMT amplifiers, could provide further improvement in the high frequency region. We expanded upon this portion of the text in the Discussion section and included a couple of references.

As for the maximum frequency in the noise PSD (1.25 MHz), this is simply the Nyquist frequency of the 2.5 MHz digitizer that was available to us for this work. This sampling rate was originally only stated in the Methods section, but we have now also included it in the Results section.